# Influence of Tumor Laterality and Focality on Clinical Implications and Tumor Advancement in Well-Differentiated Thyroid Cancer

**DOI:** 10.3390/cancers16234109

**Published:** 2024-12-07

**Authors:** Michał Miciak, Krzysztof Jurkiewicz, Anna Dziekiewicz, Szymon Biernat, Michał Kisiel, Beata Wojtczak, Dorota Diakowska, Krzysztof Kaliszewski

**Affiliations:** 1Department of General Surgery, University Centre of General and Oncological Surgery, Wroclaw Medical University, 50-556 Wrocław, Poland; krzysztof.jurkiewicz@student.umw.edu.pl (K.J.); anna.dziekiewicz@student.umw.edu.pl (A.D.); szymon.biernat@student.umw.edu.pl (S.B.); michal.kisiel@student.umw.edu.pl (M.K.); krzysztof.kaliszewski@umw.edu.pl (K.K.); 2Department of Endocrine Surgery, University Centre of General and Oncological Surgery, Wroclaw Medical University, 50-556 Wrocław, Poland; beata.wojtczak@umw.edu.pl; 3Division of Medical Biology, Faculty of Nursing and Midwifery, Wroclaw Medical University, 50-368 Wrocław, Poland; dorota.diakowska@umw.edu.pl

**Keywords:** thyroid cancer, thyroid cancer diagnostics, tumor laterality, tumor focality, thyroid surgery, thyroid cancer management

## Abstract

The diagnosis of thyroid cancer (TC) is currently supported by laboratory tests, ultrasound imaging, and fine-needle aspiration biopsy (FNAB), and surgery is one of the main treatment methods. Proper TC localization and the correct determination of its laterality and focality are important. The aforementioned features studied in this paper, as well as local and lymph node infiltration, appear to influence TC advancement. These features should be considered when selecting the therapeutic approach and surgical method. Our retrospective study of a group of 697 patients admitted to the study center revealed that unilateral and solitary TC had the lowest advancement rate and that bilateral and multifocal TC had the highest advancement rate. These findings could have important clinical implications in TC management to prevent reoperation. In future studies, researchers could verify these results and improve the management of TC patients by applying the studied preoperative features.

## 1. Introduction

The thyroid gland, a crucial endocrine organ, is exposed to many risk factors that can trigger potential diseases [1]. The most common condition worldwide is iodine deficiency, and thyroid disorders are now considered among the most common conditions world-wide and can significantly reduce the quality of life if left untreated [2]. Thyroid conditions are generally divided into functional, inflammatory, autoimmune, and neoplastic (nodules can be either benign or malignant) [3,4]. Thyroid nodules are present in up to 50% of patients and are typically benign and asymptomatic. They can manifest during the course of diseases, such as Hashimoto’s thyroiditis, subacute thyroiditis, Graves’ disease, and thyroid cancer (TC), and coexist with thyroid cysts and colloid nodules [5,6]. The incidence of TC is increasing worldwide, making it the 10th most common cancer in the world [7]. Histologically, TC can be classified into various subtypes, including papillary TC, follicular TC, medullary TC, anaplastic TC, and numerous other rarer types (e.g., lymphoma, myeloma, and sarcoma). Both papillary and follicular TCs are considered well-differentiated TCs and represent approximately 90% of all TC cases [8,9]. Well-differentiated TC is currently primarily managed with surgery, complementary radioiodine treatment, and endocrine control along with pharmacological hormonal supplementation. The procedures that are performed are total thyroidectomy or hemithyroidectomy, with the possible lymphadenectomy of the midcervical compartment [10]. On the other hand, types of TCs with a poor prognosis (such as medullary TC or anaplastic TC) require more aggressive management, especially at the start of treatment. Surgical excision remains the principal treatment for medullary TC as it is completely insensitive to radioiodine therapy. Total thyroidectomy in addition to central-only or central and lateral neck dissection is necessary to reduce the risk of recurrence and to search for involved lymph nodes. Frequent follow-up visits are also part of aggressive treatment [11]. Anaplastic TC has a poor prognosis and can be diagnosed as early as disseminated malignancy. It is characterized by the rapid infiltration of adjacent tissues, so surgical treatment may become impossible. Combinations of methods, such as surgery, radiotherapy, and chemotherapy, are needed for this type of cancer. Moreover, radiation therapy appears to be the main method of treatment for this type of TC [12]. However, well-differentiated TCs have a better prognosis and are mostly diagnosed as microcarcinomas, which, in line with the current increasing trend of cancer screening activity, may paradoxically cause their overdiagnosis and overtreatment [13,14]. The diagnosis of TC is supported by the results of physical examination, laboratory tests, ultrasonography, and fine-needle aspiration biopsy (FNAB). When a thyroid lesion is suspected, serum TSH and thyroid hormone levels should be measured to determine the functionality of the lesion because malignant lesions are rarely functional. Features on ultrasound that indicate the malignancy of a lesion include hypoechogenicity, microcalcifications, irregular borders, or a taller-than-wide shape, and when several of these features are present at once, the potential for the malignancy of the lesion increases. The definitive preoperative diagnosis relies on FNAB, which is most often performed under ultrasound guidance [15,16]. The FNAB results are graded in accordance with the Bethesda System for Reporting Thyroid Cytopathology, the current version of which was published in 2023. The system classifies lesions into six grades, with grade V indicating the presence of a lesion with a high suspicion of malignancy and grade VI indicating malignancy [17]. TC is diagnosed via postoperative histological examination and classified using the TNM Classification for Thyroid Cancer. This classification includes the location of the primary tumor (pT) in the form of tumor infiltration to the perithyroid tissues or other organs, lymph node infiltration (pN), and the presence of distant metastasis (pM) [18]. In general, the results of histopathological examination aid the preoperative diagnosis of TC and the selection of the most appropriate surgical method and follow-up protocol [19]. Moreover, preoperative diagnosis is most important as it benefits clinical decision making. As we mentioned, ultrasonographic or FNAB features reveal the presence of a lesion with a high suspicion of malignancy. It is also worth considering the location of the primary tumor, whether the tumor is bilateral or unilateral, and its single or multifocal nature. An investigation of these features is necessary for the proper management of TC and is useful for predicting TC advancement. TC that is multifocal or bilateral usually requires total thyroidectomy, with more aggressive follow-up treatment, whereas unilateral or solitary TC is not always treated appropriately as they are often overlooked because of other neoplastic foci and only identified via post-surgical histopathology [20,21,22,23]. Thus, the aim of our study was to investigate the impact of the above-mentioned features of TC laterality and focality on tumor progression. Thus, the impact of such features on tumor progression must be considered in making the correct clinical decision and selecting an appropriate treatment method with an appropriate aggressiveness reliability as such decisions affect the speed of recovery and the recurrence rate. On the basis of the studied preoperative diagnostic features and the histopathological results, clinicians can opt for a more aggressive treatment approach if necessary. Thus, owing to a thorough analysis of our findings and comprehensive review of the literature, this paper provides additional insights into the management of TC.

## 2. Materials and Methods

We retrospectively analyzed the data of 6019 patients treated at Wroclaw Medical University (University Centre of General and Oncological Surgery, Department of General Surgery). Patients underwent surgery for nodular thyroid disorders such as single thyroid nodules, multiple thyroid nodules, nodular goiters, thyroid cysts, thyroid nodules associated with Graves’ disease, and possibly TC with intraoperative nerve monitoring. The data presented were collected from January 2008 to December 2023. After the application of the inclusion and exclusion criteria (presented in Figure 1), 697 (11.6%) patients with TC were selected for further analysis.

A group of 697 (11.6%) patients with histopathologically confirmed malignant thyroid lesions (TCs) were further analyzed. In the presented group, there were 594 (85.22%) women and 103 (14.78%) men. The oldest patient was 89 years old, and the youngest was 15 years old, with a mean age of 51.4 ± 15.9 years. Patients lived in geographically iodine-sufficient regions prior to surgical intervention and were also assessed for other TC risk factors (sex, age, hereditary conditions, family history of TC, radiation exposure, and excess body weight). All patients underwent thyroid ultrasound and fine-needle aspiration biopsy (FNAB) before surgery and histopathological evaluation of the material was performed by three independent experienced pathologists after surgery. The cytological samples were classified according to the aforementioned Bethesda System for Reporting Thyroid Cytopathology and the histopathological material was classified according to the TNM classification for Thyroid Cancer [17,18]. For this classification, as positive indicators of TC advancement, we included the following: the minimal feature of pN1 (pN0: no evidence of lymph node metastasis; pN1a: metastasis to pretracheal, paratracheal, prelaryngeal or upper mediastinal lymph nodes; and pN1b: metastasis to unilateral, bilateral or contralateral lateral cervical lymph nodes or retro-pharyngeal lymph nodes) and the minimal feature of pM1 (pM0: no distant metastasis; pM1: evidence of distant metastasis). Patients underwent primary procedure of hemithyroidectomy or total thyroidectomy. The vast majority of the procedures were of total thyroidectomy, and hemithyroidectomy was only used in the following cases: smaller lesions (microcarcinomas), low-risk nature, low grade in FNAB, no extrathyroidal extension, and no clinical or radiographic evidence of lymph node metastases. Patients with TC or pathological lymph nodes also underwent therapeutic and diagnostic lymphadenectomy of the mid-cervical lymph node compartment. It should be noted that a definite diagnosis of TC is obtained by histopathological examination; therefore in a number of patients undergoing surgery for other thyroid disorders, primary prophylactic lymphadenectomy was not performed. Indications for reoperation included nodular recurrence in the other lobe and completion of thyroidectomy after diagnosis of TC on histopathological examination after hemithyroidectomy procedure. Surgical extension was also required after total thyroidectomy in case of positive imaging studies (SPECT/CT scintigraphy) during postoperative follow-up visits. All patients were evaluated, and the following data were collected: age, sex, type of surgery, need for reoperation (from patient medical records), ultrasound echogenicity of the lesion, tumor laterality, tumor focality (with the use of pre-surgical ultrasound, FNAB, and final diagnosis of these features on the basis of postoperative histopathology results), size, shape of the lesion, presence of a tumor capsule, pTNM stage, extrathyroidal extension, and vascular invasion (also from post-surgical histopathological results). The study was positively reviewed by the Institutional Review Board and Ethics Committee of Wroclaw Medical University, Wroclaw, Poland. All eligible patients provided informed consent at admission, which indicated that the results may be used for research purposes. The data were anonymized and analyzed retrospectively. The authors did not have direct access to the study participants.

### Statistical Analysis

Statistical analysis was performed using Statistica 10.0 software (StatSoft Inc., Tulsa, OK, USA). Qualitative variables were reported as numbers and percentages, and quantitative variables as medians and interquartile ranges, due to the lack of normal distribution of the data. The normality of the data distribution for quantitative variables was assessed using the Shapiro–Wilk test. Differences between groups were analyzed using Mann–Whitney U-test. Categorical variables were compared using the chi-square test. Final correlations between variables were assessed using Spearman’s correlation test, suitable for non-parametric data. A *p* value < 0.05 indicated statistical significance where appropriate.

## 3. Results

A total of 697 patients were diagnosed with TC according to the diagnostic criteria applied in this study. Widely accepted risk factors for TC were not present in most patients (622; 89.24%), and most patients underwent total thyroidectomy (501; 71.88%). Histopathological examinations revealed that most patients presented with unilateral (641; 91.97%) or solitary (510; 73.17%) lesions. The histologic types of TC were diverse, with a definite predominance of well-differentiated TCs: papillary TC (610; 87.53%) and follicular TC (33; 4.73%). Other histologic tumor types (medullary, undifferentiated, lymphoma, secondary, squamous cell, sarcoma, and myeloma) were significantly less common in the study group. Table 1 presents the characteristics of the study group and includes the 16-year observational period from 2008 to 2023.

Among all the TC patients (in the study group), 594 (85.22%) were female and 103 (14.78%) were male, with a mean age of 51.4 ± 15.9 years (16–89 years range). Most patients (622, 89.24%) did not present with the aforementioned risk factors for TC, but this difference was not statistically significant (*p* = 0.147). However, statistically significant differences were observed for the characteristics of age, tumor laterality, tumor focality, the type of surgery, the TC pTNM stage, and the previously mentioned TC histological types. As previously mentioned, most patients, 697 (91.97%), presented with unilateral TC whereas only 56 patients (8.03%) presented with bilateral TC. Differences in TC focality were also significant; 510 patients (73.17%) presented with solitary TC whereas 187 patients (28.83%) presented with multifocal TC. Most patients (501, 73.17%) underwent total thyroidectomy and 196 (28.12%) patients underwent hemithyroidectomy. With respect to TC advancement, most patients (524, 75.18%) presented with pTNM stage I disease. Significantly fewer patients had advanced disease, and a notable percentage of patients with stage III or IV disease had a more aggressive histological type of TC (mainly MTC and anaplastic TC). Consequently, we separated patients with well-differentiated TCs (papillary TC and follicular TC). Although the types of TC differ in terms of biology, histological nature, aggressiveness, and management (mainly MTC and anaplastic TC), well-differentiated TCs are the most common, presenting more indolent behavior, and similar management and prognosis, allowing us to analyze them objectively rather than in combination with other types, especially considering that MTC and anaplastic TC require a radical therapeutic approach from the beginning. Patients with well-differentiated TCs were then further divided into four groups according to the laterality and focality features of their type of TC to analyze whether combinations of these factors had a positive (or negative) effect on TC advancement (extrathyroidal extension, thyroid vascular invasion, lymph node infiltration, or presence of distant metastases, as evidenced by histopathological characteristics) and the need for reoperation (total thyroidectomy in the case of TC recurrence after initial hemithyroidectomy or radicalized thyroidectomy with the lymphadenectomy of additional cervical compartments in the case of TC recurrence after initial total thyroidectomy).

### 3.1. Unilateral and Solitary Thyroid Cancer

The records of 461 patients were analyzed (Table 2). Within this group of patients, 392 (85.03%) were female and 69 (14.97%) were male. Each of the patients in this group had a solitary lesion in only one thyroid lobe. According to the postoperative histopathology results, most patients showed the following—no extrathyroidal extension: 342 (74.19%), no vascular invasion: 339 (73.54%), no lymph node infiltration (pN0): 350 (75.92%), and no distant metastases (pM0): 399 (86.55%). In contrast, 119 patients (25.81%) presented with extrathyroidal extension, 122 patients (26.46%) presented with vascular invasion, 92 patients (19.96%) presented with positive lymph node metastases (pN1a, pN1b), and 17 patients (3.69%) presented with distant metastases. Although not always clearly obtainable, the pTNM stage was also recorded: pNx was reported for 19 patients (4.12%) and pMx was reported for 45 patients (9.76%). Among these 461 patients, 355 (77.01%) underwent total thyroidectomy and 106 (22.99%) underwent hemithyroidectomy. Most patients (382, 82.86%) did not require reoperation, but 79 patients (17.14%) did. However, it is worth noting that most patients who required reoperation underwent hemithyroidectomy. All results discussed in the table were statistically significant.

### 3.2. Unilateral and Multifocal Thyroid Cancer

The records of 128 patients were analyzed (Table 3). Within this group, 108 (84.40%) were female and 20 (15.60%) were male. The patients in this group had multifocal lesions in only one thyroid lobe. According to the postoperative histopathology results, 69 (53.91%) patients had no vascular invasion, 70 (54.69%) had no lymph node infiltration, 66 (51.57%) had pN0, and 87 (67.96%) had no distant metastasis (pM0). In contrast, 59 patients (46.09%) had extrathyroidal extension, fifty-eight (45.31%) had vascular invasion, fifty-one (39.84%) were positive for lymph node metastases (pN1a, pN1b), and five (3.91%) had distant metastases. Notably, regarding the pTNM stage, unspecified features were also recorded: pNx was reported for 11 patients (8.59%) and pMx was reported for 36 patients (28.13%). The pTNM stage is not always obtainable. Among these 128 patients, 76 patients (59.38%) underwent total thyroidectomy and 52 (40.62%) underwent hemithyroidectomy. Most patients did not require reoperation (82 patients (64.06%)), but 46 patients (35.94%) required reoperation. However, it is worth noting that most patients who required reoperation also underwent hemithyroidectomy. All results discussed in the table were statistically significant.

### 3.3. Bilateral and Solitary Thyroid Cancer

The analyzed group of patients included 14 records (Table 4). Within this group, eight (57.14%) were female and six (42.86%) were male. Patients in this group had a solitary lesion in both thyroid lobes. According to the postoperative histopathology results, more patients had extrathyroidal extension: 10 (71.43%); other results included lymph node infiltration (pN1a, pN1b): 10 (71.43%) and no distant metastases (pM0): 11 (78.56%). In contrast, for patients without extrathyroidal extension, four patients (28.57%) presented with no lymph node metastases (pN0), three patients (21.42%) presented with extrathyroidal extension, and only one patient (7.15%) presented with distant metastases (pM1). Notably, for the features resulting from the pTNM score, unspecified features were also recorded: pNx was reported for one patient (7.15%), and pMx was reported for two patients (14.29%), as these data were not always clearly obtainable. For vascular invasion, the same number of patients (seven [50,00%]) presented with no invasion or present results. Of these fourteen records, twelve patients (85.71%) underwent total thyroidectomy and two (14.29%) underwent hemithyroidectomy. Most patients did not require reoperation (twelve patients (85.71%)), but in two patients (14.29%), the procedure was needed. In this group, every patient who required reoperation underwent hemithyroidectomy. All results discussed in the table were statistically significant.

### 3.4. Bilateral and Multifocal Thyroid Cancer

The records of 40 patients were analyzed (Table 5). Within this group, thirty-two (80.00%) were female and eight (20.00%) were male. The patients in this group presented with multifocal lesions in both thyroid lobes. According to the postoperative histopathology results, 33 patients (82.50%) experienced extrathyroidal extension, 31 (77.50%) experienced vascular invasion, 33 (82.50%) experienced lymph node infiltration (pN1a, pN1b), and 28 (70.00%) had no distant metastases (pM0). Moreover, seven patients (17.50%) had no extrathyroidal extension, nine patients (22.50%) had no vascular invasion, seven patients (17.50%) had no lymph node metastases (pN0), and three patients (7.50%) had distant metastases. Notably, nine patients (8.59%) had unspecified pMx. Data regarding this feature were not always obtainable. In this group, none of the patients (0.00%) presented the pNx feature. Among these forty patients, thirty-seven patients (92.50%) underwent total thyroidectomy and three (7.50%) underwent hemithyroidectomy. Half of the patients did not require reoperation, but the other half required reoperation. However, it is worth noting that every patient who required reoperation underwent hemithyroidectomy. All results discussed in the table were statistically significant.

### 3.5. Correlation Among Bilateral, Unilateral, Solitary, and Multifocal Thyroid Cancers

Finally, an analysis of the cross-correlations between all the separate groups of TC patients was performed with respect to the applied TC-advancement-predictive features. Strong correlation was observed between the following groups: unilateral solitary–unilateral multifocal, unilateral multifocal–bilateral solitary, and bilateral solitary–bilateral multifocal. Moderate correlation was observed between the following groups: unilateral solitary–bilateral solitary, and unilateral multifocal–bilateral multifocal. Weak correlation was observed between the unilateral solitary and bilateral multifocal groups. The results are presented in Table 6. The strong and moderate correlations were indicated as relevant. The weak correlation in the last case resulted from the greatest extremity and the largest difference in the number of patients between these groups, as presented in Figure 2. In contrast, other correlations indicate that laterality and focality have implications for TC advancement.

## 4. Discussion

In this study, we presented the association of TC laterality and focality with clinical tumor advancement. The presented results and correlations between studied groups confirm both the importance of the accurate diagnosis of patients with suspected TC as well as the importance of appropriate management and surgical method selection, particularly for patients with bilateral and multifocal TC considering their increased likelihood of needing reoperation or complementary treatment such as radioiodine therapy. Histopathologically confirmed bilaterality and multifocality are considered indicators of TC advancement and taking them into account may result in a better prognosis for patients with more advanced TC.

### 4.1. Overview of the Current Thyroid Cancer Management System

Guidelines for the management of well-differentiated TC currently list various treatment approaches. These include active surveillance, hemithyroidectomy, total thyroidectomy, near thyroidectomy, and variants of lymphadenectomy in the form of central neck dissection or lateral neck dissection. They focus mainly on the tumor size; presence of extrathyroidal extension (T4), lymph node infiltration (N1), and distant metastasis (M1); and histologic type of TC (PTC or FTC) and less on laterality, focality, or vascular invasion. The guidelines are important for individualized clinical decision making [24]. It is also worth noting that TC can be indolent and asymptomatic. Thus, there is a risk of undertreatment in patients with more advanced TC, but also of overtreatment in patients with less advanced TC [25]. Recurrent laryngeal nerve palsy, hoarseness, and postoperative hypoparathyroidism are risk factors associated with advanced surgical approaches such as total thyroidectomy with lateral neck dissection [26]. The usefulness of active surveillance also remains unclear. Active surveillance is considered suitable for patients with small tumors (up to 1.0 cm or 1.5 cm, depending on the literature) and T1a- or T1b-stage disease without lymph node infiltration [27,28]. However, risk factors such as the infiltration of the trachea, infiltration of the recurrent laryngeal nerve, or advanced disease according to the Bethesda system are contraindications for active surveillance. However, active surveillance can protect patients from the aforementioned postoperative complications, but if it is not discontinued in a reasonable amount of time, it can cause clinical difficulties [29]. Surgery is recommended for more advanced TC. Hemithyroidectomy is the method of choice for patients without known risk factors for TC; no family history of the disease; unilateral, solitary lesions; no lymph node metastases; and T1a- and T1b-stage disease (including T2 when appropriate). For patients with other types of TC, total thyroidectomy remains the best method [30]. Hemithyroidectomy is not associated with a lower overall survival rate in patients with T2 stage disease but may be associated with local recurrence and the need for reoperation. However, it is important to consider the risk of complications of total thyroidectomy as the risk of complications is two times higher than that of hemithyroidectomy even when performed by an experienced surgeon [31,32]. Thus, in our study, we did not limit hemithyroidectomy to only patients with unilateral and solitary TC. Patients with bilateral or multifocal lesions who underwent only hemithyroidectomy usually require total thyroidectomy in the second lobe. According to relevant guidelines and studies, the lack of a signal on intraoperative neuromonitoring following the removal of one lobe warrants the delay of the completion of the procedure for approximately 3 months to allow the regeneration of the posterior laryngeal nerve in the event that it was injured [33]. Bilateral vocal cord paralysis is a serious complication that requires tracheostomy [34]. Importantly, it is not always possible to accurately determine focality and laterality preoperatively on the basis of ultrasound and FNAB, and the effect of these features on tumor advancement can be obtained only from postoperative histopathology results. On the basis of these findings, reoperation and total thyroidectomy can supplement the second stage of management. Molecular markers specific to TC and their role in predicting tumor advancement remain important issues for modern precision medicine. Recognized markers include genetic, miRNA, long noncoding RNA, or proteome-based markers. The most common marker is the BRAF^V600E^ mutation observed in 40–80% of PTCs [35]. It is associated with higher extrathyroidal extension, lymph node metastasis, or an advanced TC stage, so its correlation with laterality and focality also appears to be present. However, due to its heterogeneous phenotypes and the need for additional genetic combinations, this marker is not a good indicator for TC surgical treatment [36,37].

### 4.2. Influence of Laterality and Focality on Thyroid Cancer Advancement

Laterality in TC have been listed as an important risk factor for tumor advancement by numerous authors. A study of 1258 patients revealed that bilateral TC is associated with a more aggressive disease and a higher recurrence rate, thus warranting additional treatment and frequent follow-up examination. Bilaterality was shown to be significantly associated with the tumor size, infiltration of resection margins, and metastasis to lymph nodes. According to the authors of the study, up to 30% of patients presented with bilateral TC (approximately 10% of whom were in the study group), further emphasizing the importance of follow-up examinations [38]. With bilateral TC, the rates of overall survival and progression-free survival may also be reduced. This feature is also related to microscopic metastasis outside the thyroid tissue, which has been classified as a significant feature of TC aggressiveness. In our study, we also observed similar phenomena, where the majority of patients with bilateral TC showed extrathyroidal extension and a lymph node involvement of min. N1 (79.63% and 81.13%, respectively) compared to patients with unilateral TC (30.22% and 25.58%, respectively). Distant metastases are very uncommon for well-differentiated TCs, and our study revealed that distant metastases occur in in only isolated cases (we observed only 9.30% of min. pM1, even in the bilateral TC group). However, the distant metastasis of well-differentiated TCs is still possible. Al Hassan M.S. et al. presented a case of bilateral TC (exactly FTC) with retrosternal extension, lytic lesions in the sternum, and lung metastases. Treatment was aggressive and included total thyroidectomy, high-dose radioiodine treatment, and sternectomy, followed by reconstruction. Everything had to be supplemented with biologics such as levatinib owing to the resistance of the tumor and metastases to radioiodine [39]. Thus, it can be speculated that FTC is more aggressive than PTC is. There are also very rare situations in which both histological types coexist, such as in cases of unilateral TC. Consequently, a low risk of tumor progression becomes a high risk of tumor progression and therefore warrants more aggressive management [40]. The association of laterality with BRAF^V600E^ mutation has also been classified as a significant contributor to TC progression. A statistically significant correlation of the prevalence of the BRAF^V600E^ mutation was proven, where the study authors observed a higher prevalence of the mutation in bilateral TC than unilateral TC (65.7 vs. 50.4%, respectively; *p* value = 0.038) [41]. Another study also found a correlation between these features, 72.8% of patients tested with BRAF^V600E^ mutation showed bilateral TC, and additionally, the prevalence of BRAF^V600E^ mutation was higher in patients with bilateral TC than unilateral TC (63.4% vs. 42.3%, respectively; *p* value = 0.014). Therefore, it is considered that patients with detectable BRAF^V600E^ mutation may be suggested a total or near-total thyroidectomy procedure [42]. However, it is necessary to be careful about such conclusions due to the fact that the mentioned mutation in bilateral TC is not always related to other features of aggressiveness that include extrathyroidal extension or the tumor size [41].

Focality in TC is also considered a significant TC advancement factor. A large meta-analysis by Kim H. et al. revealed that multifocality was an important indicator for recurrence. The size of the primary tumor (≤ 1 cm or > 1 cm), the number of tumor foci (2 or ≥ 3), and the patient’s age were considered study variables, unlike the histological features that we found to be predictive of the tumor stage in our study. Another large meta-analysis by Joseph K.R. et al. revealed a significant association between multifocal TC and lymph node metastasis, extrathyroidal extension, or a tumor size > 1 cm (*p* values: 0.03, <0.001, and 0.88, respectively) whereas the presence of multifocality was not directly related to patient age or the male sex. We also noticed, similarly, that the majority of patients with multifocal TC showed extrathyroidal extension and a lymph node involvement of min. N1 (54.76% and 51.20%, respectively) compared to patients with solitary TC (27.16% and 21.47%, respectively). Other authors noted similar correlations; another study revealed that compared with solitarity, multifocality was characterized by significantly higher incidences of extrathyroidal extension and vascular invasion and more advanced disease, as evidenced by higher N and TNM stages, and the recurrence rate for multifocal TC was found to be statistically significant (*p* value: 0.034). The impact of present vascular invasion also in our study was in a disadvantage for multifocal TC compared to solitary TC (52.98% compared to 27.15%). However, the conclusions from the aforementioned study were similar to ours; patients with multifocal TC require careful but aggressive management, including surgery, and frequent postoperative follow-up examinations [43,44,45]. As described above, multifocal TC is intrinsically associated with aggressive histopathological features, which has been confirmed in other published studies. In cases of extrathyroidal extension and other factors mentioned by the authors (older age and female sex), total thyroidectomy should always be considered. Central neck dissection is also recommended. [46,47]. Most patients in the groups showing multifocality in our study also underwent these particular procedures (the rate of total thyroidectomy in this group was 67.2%). Indeed, it is not possible to confidently diagnose multifocal TC on the basis of preoperative examinations alone, which, as mentioned earlier, can lead to undertreatment and the use of hemithyroidectomy, which also occurred in our study (the rate of hemithyroidectomy in this group was 32.8%). However, reports in the literature indicate that multifocal TC patients who do not undergo total thyroidectomy do not have a worse prognosis than those with solitary TC [48]. Solitary TC, which, in our study had the lowest rate of node metastasis of min. pN1 (21.47%), is not recommended for prophylactic central neck dissection because of the risk of complications. However, an association between tumor localization in the upper lobe of the thyroid gland, a size > 7 cm, extrathyroidal extension, and metastasis to lateral cervical lymph nodes has been proven, and thus, such lesions warrant additional diagnostic methods and attention in clinical settings [49]. Another meta-analysis revealed that multifocal TC patients with three or more foci are prone to lymph node metastasis and tumor progression [50]. In our study, we did not focus on the number of foci, but the overall features of multifocal TC were also more closely related than those of solitary TC. Similarly, the association of distant metastasis with TC laterality and focality has been discussed in few studies, all of which were limited by a lack of access to information on pathological features that increase the risk of metastasis [51]. BRAF^V600E^ mutation has also been reported in the literature as an important factor to the focality of TC. In a study by Ahn H.Y. et al., it was observed that the group of patients with positive mutation and multifocal TC showed a higher rate of extrathyroidal invasion than the group without mutation (32.9% vs. 6.7%, *p* value = 0.041) and higher values of lymph node invasion (67.2% vs. 40.0%, *p* value = 0.049). This places the BRAF^V600E^ mutation as an aggravating factor in multifocal TC [52]. A different study observed that the mutation positively influenced not only the occurrence of multifocality relative to its absence (30.6% vs. 17%; *p* value = 0.031), but also the occurrence of distant metastases (3.1%), when none of the multifocal TCs without mutation induced them. Tumors with the mutation were also generally smaller than those without mutation (14.4 vs. 18.3 mm; *p* value = 0.018) [53]. We can conclude that patients with multifocal TC can present smaller tumors with positive BRAF^V600E^ mutations even when the largest main tumor is negative. Thus, it seems reasonable to perform the molecular testing of smaller tumors as well, which strongly emphasizes the importance of correctly determining focality in TC management [54].

Nevertheless, it is worth considering laterality together with focality as a predictive marker of TC aggressiveness. A study by Yan T. et al. revealed that bilateral multifocality, rather than unilateral multifocality, should be considered an aggressive marker of TC progression, but the authors suggested that neither feature is an independent prognostic factor and that both are best considered together [55]. Indeed, well-differentiated TCs are mostly indolent and have a good prognosis but always should be approached on a case-by-case basis, accounting for the entire clinical spectrum. Additionally, up to 46% of multifocal TC cases and 34% of bilateral multifocal TC cases are found in the pediatric population. It has been reported that only approximately one-third of multifocal TC cases and two-thirds of bilateral multifocal TC cases have been diagnosed preoperatively [56]. In certain studies, researchers identified that multifocality was a more significant contributor than bilaterality was to TC advancement; however, others drew different conclusions (a greater tendency for TC advancement and recurrence in patients with bilaterality, regard-less of multifocality) [57,58,59]. Nevertheless, in our analysis, patients with multifocality and a combination of both of these contributing features presented the highest percentages of histological factors contributing to tumor advancement, as also noted in the study by Parvathareddy S.K. et al. [60]. These features are always associated with the aggressiveness and progression of TC, but depending on the study, population, or diagnostic efficacy, they can be statistically significant independently or in combination with other factors. Laterality and focality also contribute to lymph node infiltration. In our study, we per-formed central neck dissection for diagnostic or therapeutic purposes if any lymph nodes were clinically involved. We demonstrated the correlations of the studied features including lymph node infiltration with TC advancement, so neck dissection as a means of prophylaxis may also be worth considering. Ozdemir K. et al. noted that cervical lymph node invasion was a significant risk factor in patients with bilateral multifocal TC (*p* val-ue: 0.004) and was not a significant risk factor in patients with unilateral or multifocal TC. This study also confirmed our observations, where lymph node invasion was found in as many as 82.50% of patients with bilateral multifocal TC (*p* value <0.0001), and this appeared to be by far the highest proportion in this feature. The authors reported a significant increase in the risk of metastasis (N other than N0), so prophylactic central neck dissection may be worth considering in patients with suspected bilateral multifocal TC [61]. Other features, such as male sex, extrathyroidal extension, or suspected lymph node metastasis on the unilateral tumor side, are also worth considering for prophylactic central neck dissection. The authors noted that up to 30–40% of patients with TC have cervical lymph node metastases and that even small thyroid tumors can metastasize to the cervical lymph nodes [62,63]. Our correlations revealed that few patients in the unilateral solitary TC group needed reoperation (17.14%) whereas most patients in the bilateral multifocal group (50%) were reoperated. Hemithyroidectomy instead of total thyroidectomy was associated with multifocal TC and bilateral TC, with a significant percentage of patients requiring reoperation. In patients with unilateral TC and high- or intermediate-risk factors (such as extrathyroidal extension), hemithyroidectomy may be associated with an increased risk of recurrence, but it alone provides comparable tumor control compared with full surgery, providing an opportunity for risk stratification in clinical decision making [64]. In another study, researchers clearly favored total thyroidectomy over hemithyroidectomy, especially for bilateral TC. However, the authors noted that even patients with small unilateral thyroid tumors may benefit more from total thyroidectomy because of the high risks of reoperation and recurrence [65]. In the case of solitary lesions, there is no established consensus depending on the laterality of the TC as the risk of recurrence or the risk of postoperative complications, discussed earlier, may affect the clinical decision. However, multifocal TC has a greater impact on lymph node involvement or extrathyroidal extension than does solitary TC, increasing the likelihood of total thyroidectomy. In the cited study, the authors considered hemithyroidectomy for multifocal TC without clinical evidence of lymph node metastasis, but they also identified focality as a prognostic factor for overall survival [66]. The final decision on the choice of surgery should be individualized, preceded by the best possible diagnosis and consideration of the patient’s opinion.

Our results underscore the validity of correctly determining laterality and focality in TCs both in preoperative examinations (as much as possible) and especially for postoperative histopathological examination. Correlations between the different groups confirm that laterality and focality have tremendous impacts on tumor advancement. In the available literature, many authors have concluded similarly, where bilateral TCs, multifocal TCs, and, especially, bilateral multifocal TCs are associated with greater tumor advancement in the TNM classification, greater extrathyroidal extension, a higher percentage of positive lymph node infiltration, and, usually, a higher percentage of the presence of distant metastases. The last parameter is not always supported by the studied features as it is not always obtainable and well-differentiated TCs (especially PTCs) are associated with a good prognosis; the pM1 feature is rare, which was also confirmed in our analysis. In addition, in our study, we did not focus on 5-year survival as an oncological parameter of tumor curability; rather, we focused only on the histology of the tumor itself and the resulting advancement. While this could have provided a broader view of the issues discussed, well-differentiated TCs are indolent in nature, and the aforementioned 5-year survival rate is almost 100%. In some studies, researchers avoided using the term “carcinoma” for this malignancy considering its psychological impact on the patient, but a broader discussion concerning the problem of overtreatment is beyond the scope of this paper [67]. Also, the potential variability of surgical techniques and approaches in recent years to perform prophylactic neck dissection or the use of modern surgical techniques like transoral endoscopic thyroidectomy may have oncologic safety implications and is related to the inherent need of determining tumor advancement and its laterality and focality. Prophylactic central neck dissection is recommended in criteria in advanced PTCs such as T3 or T4 and in the presence of metastasis in the lymph nodes of the lateral cervical compartment [68]. According to our results, these features correlate with bilaterality and multifocality so in case of their positivity, it is reasonable to think about prophylactic lymphadenectomy. The transoral endoscopic thyroidectomy also appear to be reasonable, oncologically safe, and not associated with insufficient lymph node excision, including the bilateral TCs [69]. In contrast, the lack of reference to these aspects derives from the retrospective nature of our study. On the other hand, based on the correlations and analysis presented, we can place laterality and focality as positive predictive values for TC advancement and the potential impact of these features on clinical guidelines. By revealing bilaterality or multifocality in a patient after total thyroidectomy, we can immediately consider complementary treatment with radioiodine and the potential need for radicalizing thyroidectomy in reoperation. The demonstration of multifocality in a patient after hemithyroidectomy can suggest the validity of this procedure, the necessity to remove the second lobe, the abandonment of “wait-and-see” treatment, and more frequent follow-up in the form of ultrasound or FNAB. Proposals for more aggressive treatment therefore seem reasonable, but it is always worth keeping in mind the possible complications after aggressive treatment, with typical examples of recurrent laryngeal nerve palsy or hypoparathyroidism. Radical treatment is associated with a higher probability of nerve damage, and performing central neck dissection with total thyroidectomy may increase the risk of hypoparathyroidism. Radioiodine therapy also carries the risk of prolonged recovery of parathyroid function [70,71,72]. Risk–benefit analysis is therefore essential in every case and should be performed by an experienced practitioner. However, introducing such a management into the guidelines would require more extensive studies on appropriately selected and larger groups of patients.

### 4.3. Study Limitations and Future Directions

This study had several limitations. First, as this was a retrospective analysis, its inherent inaccuracies could not be avoided, unlike in prospective studies. Changing guidelines in TC management over the years, as well as the approach to surgical therapy in the form of a higher proportion of active surveillance for smaller tumors, allowed hemithyroidectomy for low-risk TC, or existing debates on prophylactic lymph node neck dissection may underscore the value of laterality and focality. Second, the study was performed only in a single surgical center, and thus, the size of the sample was smaller than that in cross-sectional studies. Third, the main goal of this study was to analyze only patients with histopathologically confirmed malignant lesions who underwent thyroidectomy, which may have caused selection bias; however, the histopathology results of all patients were necessary to draw conclusions. Fourth, the numbers of patients in the last two study groups presented were considerably smaller than those in the first two, but it should be remembered that such an arrangement of thyroid lesions is relatively rare, even in larger groups of patients, and the correct detection of the studied features, especially multifocality, is not always possible. Finally, the patient groups analyzed were not subjected to molecular testing at our institution, so we also did not perform a correlation of such results with histopathological studies, which was a limitation of patient selection. In the current era of precision medicine, insights from TC molecular markers have a very high predictive value. In particular, the BRAF^V600E^ mutation, which is the most common, allows the prediction of multifocality as has been confirmed in studies. By determining this molecular marker, we can predict the mentioned feature with higher probability, which will allow more aggressive treatment from the very beginning. Thus, the more sensitive determination of bilateral multifocality can be determined on the basis of molecular studies.

Future research directions from this study could involve prospective studies with the inclusion of patients presenting unilateral, bilateral, solitary, or multifocal tumors with appropriately selected inclusion criteria in order to find out how correlations between these features affect the success of therapy aggressiveness and possible therapy adjustment during the study. Another suggestion could include prospective studies using molecular markers (BRAF^V600E^ and others) in tumors presenting accepted feature variations and determining the impact of mutations and markers on tumor advancement. We believe that the results of this study may be very useful, but the clinical situation of a particular patient should always be guided by personalized medicine.

## 5. Conclusions

This paper emphasizes the importance of accurately diagnosing TC prior to treatment, especially surgery, and the decision to implement complementary treatment should be made on the basis of confirmation of tumor progression with the help of laterality and focality. Comprehensive ultrasound examinations and biopsies evaluated by an experienced pathologist may aid the selection of the most appropriate surgical method. The demonstrated significant correlations between laterality and focality features in relation to tumor advancement underscore the possibility of applying these features as prognostic markers of choice rather the total thyroidectomy for high-risk cases and their help in clinical decision making. The final postsurgical histopathological results, which could allow for the accurate determination of the true quantity, laterality, and focality of TC lesions and their severity, could aid the selection of subsequent follow-up treatments or ascertain the need for reoperation. The results of these correlations may help clinicians make proper therapeutic approaches and improve patient outcomes. To better confirm our results, further research with imaging studies and specific criteria is needed to improve patient care.

## Figures and Tables

**Figure 1 cancers-16-04109-f001:**
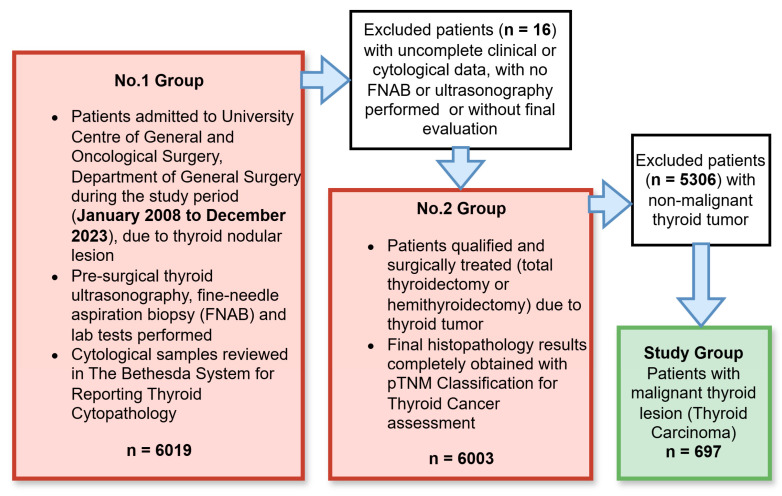
Study group selection process. A total of 6019 patients were admitted to the University Centre of General and Oncological Surgery, Department of General Surgery for thyroid disease during the observational study between January 2008 and December 2023. Patients underwent diagnostic evaluation, including fine-needle aspiration biopsy (FNAB), thyroid ultrasound examination, and laboratory tests. Patients who were eligible for surgical treatment also underwent histopathological examination to ascertain the histological type of lesion, tumor characteristics, pTNM stage, presence of thyroid vascular infiltration and thyroid capsule invasion, and presence of microcalcifications or lymph node metastases.

**Figure 2 cancers-16-04109-f002:**
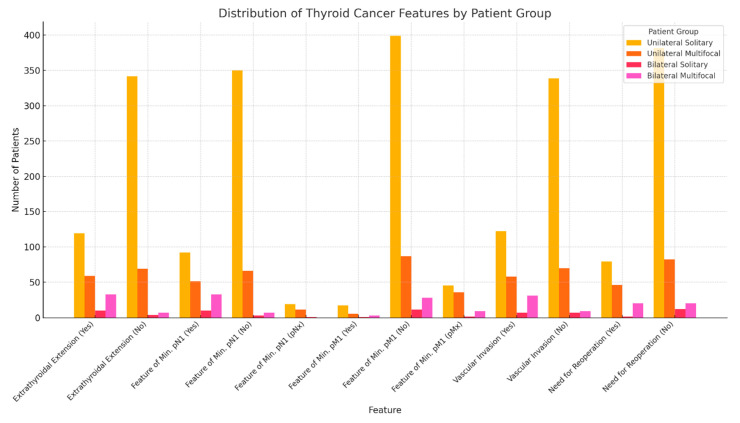
Distribution of TC features in each studied patient group in the aggregate.

**Table 1 cancers-16-04109-t001:** Characteristics of patients with TC (study group). A *p* value < 0.05 indicates a statistically significant correlation between the variables.

Feature		Number of Patients	*p* Value
Sex	Male	103 (14.78%)	0.119
	Female	594 (85.22%)	
Age (years)	<55	381 (54.66%)	<0.0001
	>55	316 (45.34%)	
TC Risk Factors	Yes	75 (10.76%)	0.147
	No	622 (89.24%)	
Tumor Laterality	Unilateral	641 (91.97%)	<0.0001
	Bilateral	56 (8.03%)	
Tumor Focality	Solitary	510 (73.17%)	<0.0001
	Multifocal	187 (28.83%)	
Type of Surgery	Total Thyroidectomy	501 (71.88%)	<0.0001
	Hemithyroidectomy	196 (28.12%)	
Tumor Histological Type	Papillary	610 (87.53%)	<0.0001
	Follicular	33 (4.73%)	
	Medullary	25 (3.59%)	
	Undifferentiated (anaplastic)	14 (2.01%)	
	Lymphoma	4 (0.57%)	
	Secondary	4 (0.57%)	
	Squamous cell	3 (0.43%)	
	Sarcoma	3 (0.43%)	
	Myeloma	1 (0.14%)	
pTNM stage	I	524 (75.18%)	<0.0001
	II	104 (14.92%)	
	III	35 (5.03%)	
	IV	34 (4.87%)	
pT	pT1a	280 (40.18%)	<0.0001
	pT1b	278 (39.89%)	
	pT2	83 (11.91%)	
	pT3	25 (3.59%)	
	pT4a	11 (1.57%)	
	pT4b	20 (2.86%)	
	pTx	0 (0.00%)	
pN	pN0	447 (64.13%)	<0.0001
	pN1a	187 (26.83%)	
	pN1b	28 (4.02%)	
	pNx	35 (5.02%)	
pM	pM0	559 (80.20%)	<0.0001
	pM1	37 (5.31%)	
	pMx	101 (14.49%)	

**Table 2 cancers-16-04109-t002:** Characteristics of patients with unilateral and solitary TC according to applied TC-advancement-predictive features. A *p* value < 0.05 indicates a statistically significant correlation between the variables.

Feature		Number of Patients	*p* Value
Extrathyroidal Extension	Present (Yes)	119 (25.81%)	<0.0001
	Absent (No)	342 (74.19%)	
Feature of Min. pN1	Present (Yes)	92 (19.96%)	<0.0001
	Absent (No)	350 (75.92%)	
	pNx	19 (4.12%)	
Feature of Min. pM1	Present (Yes)	17 (3.69%)	<0.0001
	Absent (No)	399 (86.55%)	
	pMx	45 (9.76%)	
Vascular Invasion	Present (Yes)	122 (26.46%)	<0.0001
	Absent (No)	339 (73.54%)	
Need for Reoperation	Yes	79 (17.14%)	<0.0001
	No	382 (82.86%)	

**Table 3 cancers-16-04109-t003:** Characteristics of patients with unilateral and multifocal TC according to applied TC-advancement-predictive features. A *p* value < 0.05 indicates a statistically significant correlation between the variables.

Feature		Number of Patients	*p* Value
Extrathyroidal Extension	Present (Yes)	59 (46.09%)	<0.0001
	Absent (No)	69 (53.91%)	
Feature of Min. pN1	Present (Yes)	51 (39.84%)	<0.0001
	Absent (No)	66 (51.57%)	
	pNx	11 (8.59%)	
Feature of Min. pM1	Present (Yes)	5 (3.91%)	<0.0001
	Absent (No)	87 (67.96%)	
	pMx	36 (28.13%)	
Vascular Invasion	Present (Yes)	58 (45.31%)	<0.0001
	Absent (No)	70 (54.69%)	
Need for Reoperation	Yes	46 (35.94%)	<0.0001
	No	82 (64.06%)	

**Table 4 cancers-16-04109-t004:** Characteristics of patients with bilateral and solitary TC according to applied TC-advancement-predictive features. A *p* value < 0.05 indicates a statistically significant correlation between the variables.

Feature		Number of Patients	*p* Value
Extrathyroidal Extension	Present (Yes)	10 (71.43%)	<0.0001
	Absent (No)	4 (28.57%)	
Feature of Min. pN1	Present (Yes)	10 (71.43%)	<0.0001
	Absent (No)	3 (21.42%)	
	pNx	1 (7.15%)	
Feature of Min. pM1	Present (Yes)	1 (7.15%)	<0.0001
	Absent (No)	11 (78.56%)	
	pMx	2 (14.29%)	
Vascular Invasion	Present (Yes)	7 (50.00%)	<0.0001
	Absent (No)	7 (50.00%)	
Need for Reoperation	Yes	2 (14.29%)	<0.0001
	No	12 (85.71%)	

**Table 5 cancers-16-04109-t005:** Characteristics of patients with bilateral and multifocal TC according to applied TC-advancement-predictive features. A *p* value < 0.05 indicates a statistically significant correlation between the variables.

Feature		Number of Patients	*p* Value
Extrathyroidal Extension	Present (Yes)	33 (82.50%)	<0.0001
	Absent (No)	7 (17.50%)	
Feature of Min. pN1	Present (Yes)	33 (82.50%)	<0.0001
	Absent (No)	7 (17.50%)	
	pNx	0 (0.00%)	
Feature of Min. pM1	Present (Yes)	3 (7.50%)	<0.0001
	Absent (No)	28 (70.00%)	
	pMx	9 (22.50%)	
Vascular Invasion	Present (Yes)	31 (77.50%)	<0.0001
	Absent (No)	9 (22.50%)	
Need for Reoperation	Yes	20 (50.00%)	<0.0001
	No	20 (50.00%)	

**Table 6 cancers-16-04109-t006:** Correlations among patient TC groups from the study calculated using Spearman’s correlation test. Note: values in the 0–0.3 range were assumed to indicate weak correlation, values in the 0.3–0.5 range indicated moderate correlation, and values >0.5 indicated strong correlation between groups. A *p* value < 0.05 indicates a statistically significant correlation between the variables.

	Unilateral Solitary TC	Unilateral Multifocal TC	Bilateral Solitary TC	Bilateral Multifocal TC
Unilateral Solitary TC	–	0.726 (*p* = 0.013)	0.328 (*p* = 0.035)	0.162 (*p* = 0.048)
Unilateral Multifocal TC	0.726 (*p* = 0.013)	–	0.511 (*p* = 0.024)	0.310 (*p* = 0.036)
Bilateral Solitary TC	0.328 (*p* = 0.035)	0.511 (*p* = 0.024)	–	0.580 (*p* = 0.02)
Bilateral Multifocal TC	0.162 (*p* = 0.048)	0.310 (*p* = 0.036)	0.580 (*p* = 0.02)	–

## Data Availability

The datasets used and/or analyzed during this study are available from the corresponding author upon reasonable request.

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
