# Peer review of "Influence of Tumor Laterality and Focality on Clinical Implications and Tumor Advancement in Well-Differentiated Thyroid Cancer"

_cancers, 2024, doi:10.3390/cancers16234109_

Round 1

Reviewer 1 Report

Comments and Suggestions for Authors

This study provides an in-depth analysis of how laterality and focality in thyroid cancer (TC) influence clinical outcomes and tumor progression. Here’s a detailed critique highlighting the study's strengths and suggesting potential improvements.

Strengths:

  1. Comprehensive Coverage: The study is thorough, addressing various aspects of TC management, from preoperative diagnosis to postoperative treatment recommendations and potential complications.
  2. Clear Discussion of Findings: The authors contextualize their findings within existing literature, which strengthens the study’s relevance and shows awareness of different perspectives on TC management.
  3. Evidence-Based Recommendations: The study uses findings to suggest practical, evidence-based recommendations on when to consider aggressive treatment options, such as total thyroidectomy or prophylactic neck dissection.

Areas for Improvement:

  1. Clarity and Structure:
    • Repetition and Redundancy: There is considerable repetition, especially in discussing multifocal and bilateral TC. Reiterating points makes the narrative hard to follow and detracts from the study's key findings.
    • Organizational Flow: The structure could be improved by grouping related sections (e.g., discussing bilateral and multifocal TC in one cohesive segment, rather than interspersed throughout). This would improve readability and help the reader follow the progression of arguments.
  2. Expanded Focus on Limitations:
    • Selection Bias and Molecular Testing: The limitations of not performing molecular testing and the potential selection bias should be elaborated upon. For example, clarifying how lack of molecular insights might influence the study’s applicability in precision medicine settings would give readers a clearer picture of the study's generalizability.
    • Retrospective Nature: While the study mentions retrospective limitations, discussing specific inherent biases—such as potential variability in surgical techniques or diagnostic criteria across years—would add transparency.
  3. Interpretation of Results:
    • Prognostic Value of Laterality and Focality: While the study highlights laterality and focality as factors in tumor progression, it does not fully explain how these insights might shift current clinical decision-making. Would these factors suggest adjustments in staging criteria, or would they serve as additional considerations in high-risk cases?
    • Risk-Benefit Analysis: When recommending aggressive treatments, the study could delve further into balancing these recommendations with the potential for complications (e.g., recurrent laryngeal nerve injury or hypoparathyroidism). Including a more nuanced discussion on this trade-off would offer practical guidance to clinicians.
  4. Literature Contextualization:
    • Comparative Analysis: While the study references multiple other studies on TC, a comparative analysis of the study's findings with these past studies would add value. This could be achieved by briefly summarizing how this study’s results either corroborate or differ from previous research.
    • Consideration of Molecular Markers: Although this study did not include molecular data, it might still benefit from discussing how molecular markers (like BRAF mutations) are known to correlate with focality, laterality, or aggressiveness in TC. This would situate the study within the broader, evolving field of molecular diagnostics in TC.
  5. Concluding Remarks:
    • Impact on Clinical Guidelines: The study suggests that laterality and focality should influence treatment planning but stops short of recommending specific changes to existing guidelines. Proposing how these factors might be incorporated into guidelines—perhaps as risk stratification tools for more aggressive monitoring or surgical interventions—would add a stronger, more actionable conclusion.
    • Future Research Directions: The study would benefit from suggestions on future research, such as prospective studies to validate these findings or studies that explore integrating molecular markers with focality and laterality. This would highlight the study’s implications for ongoing research and development in TC management.

Conclusion Improvement:

The conclusion could benefit from emphasizing the potential for laterality and focality to serve as prognostic markers, particularly in high-risk cases. Highlighting how these factors might better inform clinical decision-making, such as choices between hemithyroidectomy and total thyroidectomy, would make the findings more applicable to practitioners.

Reviewer 2 Report

Comments and Suggestions for Authors

Good design study.

In the method the authors should clearly describe the operation strategy in their institutes

for both primary operations and re-operation .

1) extension of surgical field:thyroidectomy & lymph node dissection.

2) indication for re-operation and surgical extension

Define the the meaning Min for MinpN and MinpM

Reviewer 3 Report

Comments and Suggestions for Authors

Dear authors,

Your study, “Influence of Tumour Laterality and Focality on Clinical Implications and Tumour Advancement in Well-Differentiated Thyroid Cancer”, cancers-3321673, is written very nicely, the English is fine, and the sentences are easy to follow. It includes a large number of patients (in terms of thyroid cancer research), which is also a benefit. But unfortunately, it has some major flows: the manuscript only describes the clinical data of TC patients, it has no novelty, and the statistical analysis is not performed well, and therefore your conclusions are not supported by your results.

In Section 2.1. it is only written that qualitative variables were compared between groups using the chi-square test and Student's t test where appropriate. First of all, what about quantitative variables, and did you test the distribution of your data? A t-test is a statistical test for testing how significant the difference is between the 'means' of 2 groups (therefore, it assumes normal distribution of the data). A chi-square test is a statistical test for categorical data, and it is used in the analysis of contingency tables (the minimum is table 2x2). Furthermore, as I could have noticed, all the tables in the manuscript only present the correlations (not the associations, nor do they test the difference), and the test of correlation is not mentioned in Section 2.1. Therefore, please rewrite the section.

Second, if you perform only the test of correlation, you should paraphrase your conclusions.

Third, please explain the sentence (that is repeated in the text): “All results discussed in the table were considered statistically significant (p < 0.0001).” What does it mean “considered” in the results section? Was the test applied, or was it an assumption? If the appropriate statistical test was performed, and if the p<0.05 (or some other set value) was gained, then the results are statistically significant.

For the improvement of your work, I suggest you add a new section in which you would merge the data from tables 2, 3, 4, and 5. Then, apply the appropriate statistical test for testing the difference in the distribution and median/average value between the groups. If you gain the statistically significant difference in such comparison, you may conclude that tumor laterality and focality have implications for tumor advancement in WDTC.

Beside these major remarks, here are some minor:

  • The summary and abstract should incorporate the number of TC patients included in the study (697).
  • As all the tables in the manuscript represent the results of Pearson’s correlation test, then the value of the correlation coefficient (r) should also be presented in the tables.
  • Table 1: Tumor laterality, focality, and type of surgery are the features that should be tested separately, so the appropriate p-value should be presented for each characteristic tested.
  • Why is the p-value for tumor histological type in Table 1 missing?
  • Table 2 and Figure 2, Table 3 and Figure 3, Table 4 and Figure 4, Table 5 and Figure 5 present the same data. There is no need for both. Therefore, one of them (tables or figures) should be excluded.
  • The titles of the figures are incorrect. There is no analysis presented in the figures. The figures are presenting only the number of patients in each category. Please correct properly.

Round 2

Reviewer 3 Report

Comments and Suggestions for Authors

Dear Authors,

Your manuscript “Influence of Tumour Laterality and Focality on Clinical Implications and Tumour Advancement in Well-Differentiated Thyroid Cancer”, cancers-3321673, has been sufficiently improved, and I recommend it publication after a few minor comments:

In the Answer to Reviewers it is said that “The data show that in the vast majority of cases the null hypothesis is rejected, and the presented patients do not show a normal distribution,…”. If so, all the applied statistical tests should be nonparametric. Therefore, please, apply the appropriate statistical test (e.g. Spearman’s correlation instead of Pearson’s correlation). In addition, the value of the p-value should be incorporated into Table 6.
